# Exploring heterogeneity in treatment effects: The impact and interaction of asset-based wealth and mass azithromycin distribution on child mortality

Elisabeth A. Gebreegziabher[1,2]*, Ali Sié[3], Mamadou Ouattara[3], Mamadou Bountogo[3], Boubacar Coulibaly[3], Valentin Boudo[3], Thierry Ouedraogo[3], Elodie Lebas[1], Huiyu Hu[1], Pearl Anne Ante-Testard[1], Steven E. Gregorich[4], Kieran S. O'Brien[1,5], Michelle S. Hsiang[2,6,7], David V. Glidden[2], Benjamin F. Arnold[1,5,6], Thomas M. Lietman[1,2,5,6], Catherine E. Oldenburg[1,2,5,6]*

1 Francis I. Proctor Foundation, University of California San Francisco, San Francisco, California, United States of America, 2 Department of Epidemiology and Biostatistics, University of California, San Francisco, United States of America, 3 Centre de Recherche en Sante de Nouna, Nouna, Burkina Faso, 4 Department of Medicine, University of California, San Francisco, California, United States of America, 5 Department of Ophthalmology, University of California, San Francisco, California, United States of America, 6 Institute for Global Health Sciences, University of California, San Francisco, United States of America, 7 Department of Pediatrics, Division of Pediatric Infectious Diseases, UCSF, United States of America

* catherine.oldenburg@ucsf.edu (CEO); amareelisabeth@gmail.com (EG)

## Abstract

### Objective

To examine how child mortality among children aged 1–59 months varies by asset-based wealth status in rural Burkina Faso, and to assess the interaction between mass azithromycin (AZ) distribution and wealth status on child mortality at both the household and community levels.

### Methods

We used data from a cluster-randomized trial and population census data on household characteristics and assets. A wealth index score for each household, used to classify the population by wealth, was generated using principal component analysis. We used the Relative Index of Inequality (RII), the Slope Index of Inequality (SII), and the concentration index to assess wealth-related inequalities in mortality, and the Gini Index to assess variability in child mortality across households and communities. Poisson regression models were used, with person-time at risk included as an offset, and robust standard error to estimate changes in mortality rates by wealth and treatment arm. Interaction was assessed on both the multiplicative and additive scales.

**Data availability statement:** The datasets analyzed during the current study are available in the Open Science Framework (OSF) repository at https://doi.org/10.17605/OSF.IO/RTX3K.

**Funding:** The CHAT trial was supported by the Gates Foundation (grant number OPP1187628). Research reported in this manuscript was also supported by the National Institutes of Health Eunice Kennedy Shriver National Institute of Child Health & Human Development (NIH/NICHD) F31 Award (1F31HD114434-01A1: E.A.G.).

**Competing interests:** NO authors have competing interests.

## Results

Mortality declined with increasing wealth at both the household and community levels, with a significant gradient at the community level (RII = 1.17, 95% CI: 1.05–1.29; SII = 2.3 per 1,000 person-years, 95% CI: 0.2–4.4), reflecting higher mortality among the poorest. The effect of AZ did not vary significantly by wealth index, and changes in mortality rates across wealth levels were similar between the two treatment arms. There was no evidence of a statistically significant interaction between AZ and asset-based wealth on either the multiplicative or additive scale at the household or cluster level.

## Conclusion

Our findings demonstrate a wealth gradient in child mortality, with the highest mortality rates observed among households and communities in the lowest wealth quintiles. These disparities were consistent across both AZ-treated and placebo groups, suggesting that the role of AZ in health disparities may be limited to addressing gaps in treatment access rather than broader wealth-related disparities. While the study may have been underpowered to detect modest interaction effects, AZ appeared to offer similar benefits across economically diverse communities, with no evidence suggesting enhanced benefits for disadvantaged communities or for prioritizing treatment based on wealth status. Further work is needed to address the wealth-related disparities in child mortality in these communities.

## Trial registration

ClinicalTrials.gov NCT03676764

## Introduction

The Sustainable Development Goals aim to end preventable deaths of children under 5 by 2030 [1]. Although significant progress has been made in reducing global child mortality, continued improvements in child survival remain a global priority [2]. As of 2022, the global under-five mortality rate was 37 deaths per 1000 live births, a significant decline from the 93 deaths per 1,000 live births in 1990 [2]. However, there is considerable variability in mortality rates and improvements across regions [3], with substantial inequalities observed in sub-Saharan Africa [4]. For instance, in 2019, two regions accounted for the majority of under-5 deaths: sub-Saharan Africa with 55% (53–57) of global under-5 deaths, and south Asia with 26% (26–27) of the total [5]. Considering such disparities, assessing local and regional trends can help identify areas for improvement despite overall progress, and help prioritize child survival interventions for those who need them most [3]. Understanding context-specific differences is essential for evaluating health equity, informing policy decisions, and effectively addressing local health disparities and their specific challenges.

The leading causes of child mortality, particularly in sub-Saharan Africa, include infections such as malaria, diarrhea, and pneumonia [6]. Addressing these issues involves strengthening health systems, implementing nutritional interventions, improving water, sanitation, and hygiene services, and preventing infections [7]. Antibiotic-based strategies, such as mass distribution of azithromycin (AZ), have also been used to reduce infection-related child mortality in these settings. AZ was selected for mass-distribution trials based on strong prior evidence from large randomized controlled trials, including the MORDOR study in Malawi, Niger, and Tanzania, which demonstrated significant reductions in all-cause childhood mortality following community-wide AZ administration [8]. These results led to further investigations, including the CHAT trial in Burkina Faso and the AVENIR trial in Niger, to evaluate whether these benefits could be replicated in other populations and contexts [9–11]. The favorable pharmacologic properties of AZ including its long half-life, broad antimicrobial activity against respiratory and enteric pathogens, and well-established safety profile, make it suitable for large-scale community interventions [12].

Previous evidence suggests that AZ is not only effective in reducing infection-related child mortality but can also help buffer against disparities in child mortality by potentially addressing gaps in treatment [13]. A secondary analysis of the MORDOR trial found that the effect of mass drug administration (MDA) with AZ was larger in communities farther from clinics [13]. Since distance from health facilities is a barrier to accessing routine and life-saving interventions [14,15], hard-to-reach and rural communities, which often lack resources [15,16], may benefit more from mass AZ treatment. Another key determinant of access to treatment and care is socioeconomic status (SES) [17]. Socioeconomic factors are known to be important determinants of child mortality, by which disparities in health outcomes have been previously noted [4,18]. Individuals in poorer communities often have less access to quality care, services and information [17], and may therefore also benefit more from mass AZ treatment. However, despite its suggested effect in addressing gaps in treatment, there is no clear evidence whether MDA of AZ interacts with wealth status to help reduce wealth-related inequalities in child mortality.

Additionally, poorer communities may have greater exposure to infectious diseases due to crowding, poorer housing structures, and limited access to clean water and sanitation [19]. Consequently, the poorest communities may experience a higher infection burden and benefit more (have more harm averted) from treatment [20]. For instance, previous research shows that reducing childhood infections, particularly in families with lower SES, may more substantially reduce the burden of cardiovascular disease in adults compared to families with higher SES [21].

Therefore, examining whether the effect of AZ varies by wealth level may help determine whether AZ MDA is more beneficial for households and communities with lower SES, which typically have reduced access to resources, healthcare, and antibiotics, and aid in resource prioritization and the development of targeted approaches.

Evaluating these effects at both the household and cluster/community levels may provide a comprehensive understanding of how MDA interventions could affect wealth-related inequalities. Since socioeconomic determinants operate at multiple levels such as individual, household, and community [22], this approach may provide insights into effects within individual households, as well as broader patterns and resource gaps at the community level, enabling more targeted strategies.

Using data from a cluster randomized trial of 278 communities in Burkina Faso [9] (CHAT) and data on household characteristics, we examined whether there is an asset-based wealth gradient in under-five mortality in these communities and whether there is an interaction between mass AZ distribution and wealth status on child mortality.

## Methods

### Study design, setting, and population

This secondary analysis utilized data from the CHAT trial (ClinicalTrials.gov, NCT03676764) and from a pre-census survey collected prior to the first CHAT census, before the trial began. The CHAT trial was a cluster-randomized study aimed

at assessing the effectiveness of mass AZ distribution for prevention of infection-related mortality in children aged 1–59 months [23]. In the CHAT trial, azithromycin was distributed biannually to children as part of a community-based intervention aimed at reducing childhood mortality. Although azithromycin is an antibiotic primarily used to treat bacterial infections, its potential mortality benefit in mass drug administration programs has been attributed to its broad antimicrobial activity, including against respiratory and enteric pathogens common in low-resource settings [24]. The trial was conducted from August 2019 to February 2023 and involved children under five in the Nouna District of Burkina Faso, including both the Nouna Health and Demographic Surveillance Site (HDSS) and the surrounding non-HDSS area [23].

Under-five mortality in Burkina Faso declined substantially [25], from 184 per 1,000 live births in 2003–48 per 1,000 live births in 2021, largely due to improvements in prevention and care [1]. Malaria, pneumonia, and diarrhea remain the leading causes of child death [26]. A previous study found that the median rate of visits to government-run primary healthcare facilities for children under five in the communities was 6.7 per 100 child-months, with most visits due to pneumonia (37.5%), malaria (25.1%), and diarrhea (9.1%) [15]. The median distance to a healthcare facility for a child was approximately 5 km [15]. Over 40% of the population of Burkina Faso lives below the poverty line [27], with poverty concentrated in rural areas [28].

## Data collection

In the double-masked, placebo-controlled CHAT trial, MDA of AZ or placebo was assigned at the cluster/community level. Eligible children in treatment clusters received oral azithromycin twice yearly, while those in placebo clusters received oral placebo on the same schedule over a period of three years (2019–2023; 6 treatment distributions in total) [23]. A census was conducted every 6 months to record births, deaths, pregnancies, and migrations. The study employed an open cohort design, with varying person-time contributions from enrolled children as they could age in or out of the cohort, migrate or die. Vital status updates for each child, along with their household and community identifiers, were recorded during each 6-month census phase. Over the study period, six rounds of census and treatment were performed, and a total of 1,086 deaths were recorded across 119,139 person-years [9]. Detailed methods of the CHAT trial have been described previously [9,23].

The CHAT trial was reviewed and approved by the Institutional Review Boards at the University of California, San Francisco, and the Comité National d'Ethique pour la Recherche (National Ethics Committee of Burkina Faso) in Ouagadougou, Burkina Faso. Written informed consent was obtained from the parent or legal guardian of each enrolled child.

Before the study began, the study team conducted a pre-census survey in study communities, collecting data on household-level wealth indicators in regions outside of the HDSS. These data included habitat type (e.g., simple detached house, villa, multiple-dwelling building, apartment building), ownership status (e.g., owner, tenant, housed by employer/parents/friends, other), and housing structure, such as the construction material used for walls (hard, semi-hard, banco, straw), floors (tiles, cement, clay, sand), and roofs (concrete, tiles, sheets, straw/leaf, crammed earth). It also included the main source of cooking energy (electricity, gas, coal/wood, petroleum stove) and access to clean water and toilets (e.g., type of toilet such as flush toilet, latrines, unconfined latrines, no toilet) and the main source of drinking water (faucet, well, drilling, river). Data on asset ownership was also collected, including items such as radio, television, video/DVD player, telephone, freezer, gas cooker, generator, computer, car, motor, and tricycle. The pre-census data covered 15,291 households in 170 clusters involved in the CHAT trial, while 12,872 households in CHAT were not included in the pre-census.

## Statistical analysis methods

We conducted three steps: 1) generated the wealth index score for each household, 2) assessed how child mortality rates change with the asset-based wealth index, and 3) evaluated the interaction between the azithromycin MDA intervention and wealth on all-cause mortality. The latter two analyses were conducted at both the household and cluster levels. Descriptive analyses were used to describe the characteristics of households from the pre-census, by treatment arm.

While the overall sample size was based on childhood mortality for the AZ versus placebo comparison in CHAT trial [9], only households present in both the pre-census and the trial were included in the household-level analyses. Households with missing pre-census wealth data were excluded from the household-level analysis using listwise deletion (i.e., complete case analysis). For cluster-level analyses, clusters with at least one household in the pre-census were included in the cluster level analyses.

## Principal component analysis (PCA)

The wealth index score for each household was created using PCA, which is a data reduction technique that reduces dimensionality by replacing many correlated variables with a smaller set of uncorrelated 'principal components' that explain most of the variance. [29,30]. PCA was used to generate the wealth index because it efficiently summarizes correlated asset variables and is widely applied in demographic and health studies [31,32]. To prepare variables for PCA, household characteristics were recoded as either improved or unimproved based on DHS categorization [31]. Ownership of assets was recorded as binary. We assessed household characteristics, their correlations, eigenvalues and the factors that explained majority of the variability.

Of these characteristics/assets, 14 were selected for inclusion in the PCA based on higher prevalence, component loading, and their positive association with wealth status. These included improved wall, floor, roof, and toilet, as well as ownership of radio, TV, video/DVD, mobile, freezer, solar plate, solar lamp, motorcycle, bicycle and cart. We used the first component that explained the largest proportion of the total variance to create the wealth score, with higher scores representing wealthier households.

We categorized the wealth index into quintiles, dividing households into five groups from the least wealthy 20% to the wealthiest 20% based on this relative measure of poverty [33].

## Wealth status and mortality

The wealth index score for each household was merged with the mortality data, which included the number of deaths, person-time at risk (aggregated by household), and the corresponding treatment value (AZ vs placebo) for the cluster in which each household was located. We examined whether mortality rate in children under five changes with increase in each quintile of the wealth index. Quintiles were analyzed both as a linear ordinal predictor and as categorical variables (with the least wealthy group as the reference), following tests for linear trends across categories.

We used a Poisson regression model with child mortality as the outcome, wealth index as the main predictor, person-years as an offset, a log link, and robust standard errors clustered at the level of randomization (i.e., community/cluster) to account for the cluster-level treatment in the household-level analysis. The margins command in Stata was used to estimate incidence rate differences (IRDs) with 95% CIs (marginal effects) using the delta method. In the adjusted analysis, we included distance to facility as a covariate, since it has been previously identified as strong predictor of child mortality in this setting [34], and is often associated with wealth [35].

We also assessed wealth-related inequalities in mortality by calculating the Relative Index of Inequality (RII) and the Slope Index of Inequality (SII) which are key measures used in epidemiologic studies to quantify and compare health inequalities across populations [36]. The RII represents the ratio of predicted outcomes between the wealthiest and poorest quintiles in the wealth distribution of the population, whereas the SII reflects the absolute difference between these groups [36]. RII and SII values of 1 and 0 indicate no inequality, respectively, while higher values indicate worse outcomes for the most disadvantaged group. We used bootstrapping with 1,000 replicates, resampling clusters with replacement, to calculate the 95% confidence limits for these estimates. Additionally, we used the Gini index and the concentration index. The Gini index, commonly used to measure income inequality, is also used to measure health inequality by providing estimates that capture the distribution of health or health risks [37,38]. We used the Gini index to assess the overall distribution and degree of inequality in child mortality across households and communities. The Gini index ranges from 0

to 1, where a value of 0 represents perfect equality and a value of 1 indicates perfect inequality [38]. The concentration index captures the extent to which health outcomes differ across individuals or communities ranked by an indicator of socioeconomic status, and is commonly used to measure socioeconomic related health inequality [39]. The concentration index was used to measure how child mortality is concentrated among different wealth groups. It is defined as twice the area between the concentration curve and the 45° line, which represents equality, and ranges from −1–1. For an ill-health outcome (such as child mortality), a negative concentration index indicates that ill health is higher among the poor. A value of −1 represents maximal pro-rich inequality (disproportionate burden on the poor), 0 indicates no inequality, and 1 represents maximal pro-poor inequality (disproportionate burden on the wealthy) [39,40].

### Wealth-AZ interaction

We examined whether the effect of AZ on child mortality varies by asset-based wealth and whether the relationship between wealth and mortality varies by treatment arm. We used the interaction directed acyclic graph (DAG) [41] shown in Figs 1A and 1B to determine and visualize the relationships between variables relevant for the interaction analyses. We used Poisson regression models similar to those described above, including both the main effects of wealth and AZ, as well as their interaction term to assess interaction on a multiplicative scale. To evaluate interaction on an additive scale, we calculated the relative excess risk due to interaction (RERI) with bootstrap 95% confidence intervals, resampling clusters with replacement, using 1000 repetitions. Based on the interaction DAG, distance to facility was included as a covariate in adjusted models. For the household-level interaction analysis, a mixed-effects Poisson model with a random intercept at the cluster level was used as a sensitivity analysis to assess the robustness of the interaction results.

In addition to household-level analyses, we conducted these analyses at the cluster level by aggregating deaths and person-time by cluster and averaging household wealth scores to determine wealth quintiles for each cluster. This approach provided insights into community-level relationships and allowed the use of pre-census household wealth data to calculate the wealth index for clusters with at least one household in the pre-census. As a result, clusters with at least one household in the pre-census were included in the cluster-level analysis, enabling the inclusion of mortality data from clusters missing households in the pre-census data.

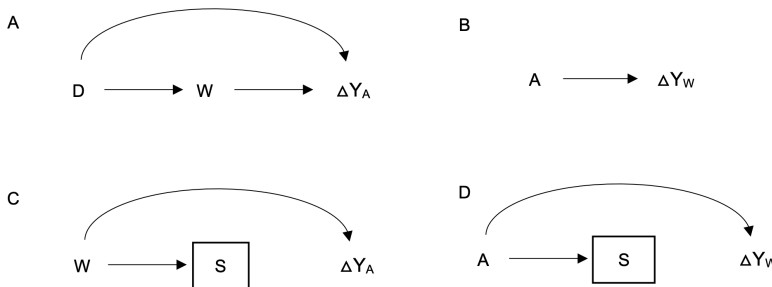

**Note:** W- Wealth, A- AZ treatment, S-Selection (household being included in pre-census), D-Distance to health facility, Y-under-five mortality, ΔY$_A$-effect of AZ on mortality, ΔY$_W$- change in mortality by wealth. Figure 1A-how wealth may modify the effect of AZ on mortality with distance to health facility as a potential confounder; Figure 1B-how AZ may influence the change in mortality by wealth; Figure 1C- selection as a threat to generalizability to target population -sensitivity analysis examined association between selection (S) and effect of AZ on mortality (ΔY$_A$). Figure 1D- selection as a threat to generalizability to target population, sensitivity analysis examined association between S (selection) and A (AZ)

**Fig 1. Interaction directed acyclic graphs illustrating the interaction between mass AZ treatment and asset-based wealth.**

## Sensitivity analyses

We conducted several sensitivity analyses. First, since the first 4 principal components explained a much larger proportion of the total variance than just the first component, we combined them (using weighted sum) to create the wealth score, which was then used in analyses similar to those conducted for the wealth index constructed from the first component [29]. Second, because households not included in the pre-census were excluded from the household-level analysis, we examined whether the effect of AZ on mortality varied by selection/inclusion into the analysis (Fig 1C). Third, we assessed whether selection/inclusion into the household-level analyses varied by treatment arm to determine if changes in mortality by wealth could be related to selection through treatment (Fig 1D). The latter two sensitivity analyses were conducted to assess potential selection bias and determine whether the results from the selected population can be generalized to the full sample and target population [41].

SAS 9.4 (SAS Institute, Cary, NC) was used for data cleaning and for generating the datasets for analyses. Stata version 14.2 (StataCorp, College Station, TX) was used for all other analyses.

## Results

There were 15,291 households in 170 clusters included in the analysis. About 53% of these households were in communities randomized to receive AZ (Table 1). The overall coverage of MDA was approximately 90% for both AZ and placebo arms. The proportion of communities that received AZ in each wealth quintile was: 58.8% (quintile 1, poorest), 47.1% (quintile 2), 58.8% (quintile 3), 50.0% (quintile 4), and 50.0% (quintile 5, wealthiest). Over 97% of participants in both arms lived in self-owned habitats, and over 75% resided in simple detached houses or villas. Housing materials varied: while three-quarters of households had relatively improved roof materials (concrete, tiles, or sheets), less than 40% had improved floor materials (tiles or cement), and fewer than 10% had houses with improved wall materials (hard or semi-hard walls). 5.3% of households in the AZ group and 4.5% in the placebo group had access to flush toilets or latrines. In both arms, less than 1% had access to clean drinking water, and similarly low percentages used electricity or gas for cooking. The most commonly owned assets, with over 70% prevalence in both groups, included mobile phones, bicycles, and solar lamps. The largest differences between the groups were in the ownership of radios and solar plates, with 44.6% and 57.5% in the AZ group compared to 49.3% and 61.5% in the placebo group, respectively.

## PCA

In the PCA, the first four components had eigenvalues greater than 1, explaining 46% of the variability, with the first component accounting for the largest portion at 20.5%. The Kaiser-Meyer-Olkin (KMO) Measure of Sampling Adequacy was 0.78, indicating that the data was suitable for PCA [42].

Ownership of solar plates, motorbikes, radios, and TVs, as well as having an improved floor, had component loadings greater than 0.3, indicating a larger contribution to the asset score. Approximately 99% of the standardized asset score values ranged from −3.5 to 4.2, with a maximum score of 6.2, and a mean of 2.6. Based on the asset score rankings, the wealth index had five levels, with the least wealthy 20% of households in the 1st quintile and the wealthiest 20% in the 5th quintile.

## Wealth and mortality

The mortality rate across all households was 9.7 per 1,000 person-years, 95% CI: 8.3 to 11.2 (Table 2). Mortality rates varied by wealth, with households in the first quintile experiencing the highest rate (12.3 per 1,000 person-years, 95% CI: 9.4 to 15.2) and those in the fifth quintile having the lowest rate (7.0 per 1,000 person-years, 95% CI: 5.3 to 8.6). There was a 11% decrease in mortality rate for each one-level increase in wealth quintile (IRR = 0.89, 95% CI: 0.83 to 0.95). With measures of inequality, on the relative scale (RII), children in households in the lowest wealth quintile were 1.12 times (95% CI: 1.05 to 1.19) more likely to die than

**Table 1. Characteristics of households from the pre-census (n = 15, 291 households in 170 clusters).**

| | Azithromycin (n = 8,157 households) | Placebo (n = 7,134 households) |
|---|---|---|
| Persons per household (mean, SD) | 7.24 (4.86) | 7.17 (4.86) |
| Children per household (mean, SD) | 1.59 (1.39) | 1.56 (1.37) |
| **Household characteristics (n, %)** | | |
| Improved housing (case, simple detached house, villa) | 6254 (76.7%) | 5522 (77.4%) |
| Ownership of habitat (own) | 7983 (97.9%) | 7027 (98.5%) |
| Improved wall material (hard, semi-hard)[†] | 535 (6.6%) | 594 (8.3%) |
| Improved floor material (tiles, cement)[†] | 2917 (35.8%) | 2573 (36.1%) |
| Improved roof material (concrete, tiles, sheets)[†] | 6201 (76%) | 5343 (74.9%) |
| Source of cooking energy (electricity, gas) | 43 (0.5%) | 18 (0.3%) |
| Access to toilet (flush toilet, latrines)[†] | 430 (5.3%) | 329 (4.6%) |
| Access to clean drinking water (source-faucet, well) | 52 (0.6%) | 41 (0.6%) |
| **Ownership of assets** | | |
| Radio[†] | 3593 (44.6%) | 3503 (49.3%) |
| TV[†] | 991 (12.6%) | 884 (12.7%) |
| Video/ CD/ DVD Player[†] | 262 (3.4%) | 250 (3.6%) |
| Fixed telephone | 11 (0.1%) | 14 (0.2%) |
| Mobile telephone[†] | 6113 (75.4%) | 5559 (78.4%) |
| Freezer[†] | 27 (0.3%) | 5 (0.1%) |
| Solar plate[†] | 4645 (57.5%) | 4362 (61.5%) |
| Generator | 7 (0.1%) | 4 (0.1%) |
| Solar lamp[†] | 5951 (73.9%) | 5426 (77.4%) |
| Petroleum lamp | 8 (0.1%) | 16 (0.2%) |
| Car | 14 (0.2%) | 8 (0.1%) |
| Motorbike[†] | 3254 (40.9%) | 3066 (43.7%) |
| Tricycle | 87 (1.1%) | 70 (1%) |
| Bicycle[†] | 5792 (71.6%) | 5288 (74.6%) |
| Cart[†] | 4695 (58.2%) | 4415 (62.4%) |
| Computer office | 3 (0%) | 0 (0%) |
| Laptop computer | 14 (0.2%) | 8 (0.1%) |

Note: [†]Characteristics included in principal component analysis.

those in the wealthiest households. On the absolute scale, households in the lowest socioeconomic status experienced approximately 1.5 additional child deaths per 1,000 person-years (95% CI: 0.5 to 2.5) compared to the wealthiest households (Table 3). The Gini index was 0.59 (95% CI: 0.56 to 0.62) at the household level and 0.40 (95% CI: 0.37 to 0.43) at the community level, indicating high inequality in child mortality within households, with some households experiencing much higher child mortality rates than others, and moderate inequality across communities (Table 3). The Concentration Index was −0.14 (95% CI: −0.25 to −0.04) at the household level and −0.15 (95% CI: −0.24 to −0.06) at the community level, reflecting modest but statistically significant inequality in child mortality by wealth, disadvantaging the poor (Fig 2).

Mortality rates decreased with increasing wealth quintiles at both the household and cluster levels, with the greater reductions occurring from the first to the second and the fourth to the fifth quintile (Fig 3). Although the trend was similar,

**Table 2. Mortality rates, rate ratios, and rate differences by asset-based wealth quintiles.**

| | Mortality Rate per 1000 PY | Unadjusted | | Adjusted for distance to facility | |
| --- | --- | --- | --- | --- | --- |
| | | IRR | IRD per 1000 PY | IRR | IRD per 1000 PY |
| **Household level** | Across quintiles | | | | |
| Wealth index quintile linear | 9.7 (8.3 to 11.2) | 0.89 (0.83 to 0.95) | −1.1 (−1.8 to −0.5) | 0.89 (0.83 to 0.95) | −1.2 (−1.8 to −0.5) |
| Wealth index quintile | In each quintile | | | | |
| 1st (least wealthy) | 12.3 (9.4 to 15.2) | ref | ref | ref | ref |
| 2nd | 10.1 (8.1 to 12.2) | 0.83 (0.65 to 1.05) | −2.1 (−4.9 to 0.6) | 0.82 (0.65 to 1.05) | −2.2 (−5 to 0.6) |
| 3rd | 9.6 (7.6 to 11.6) | 0.78 (0.6 to 1.01) | −2.7 (−5.7 to 0.3) | 0.78 (0.6 to 1.01) | −2.8 (−5.7 to 0.2) |
| 4th | 9.5 (7.4 to 11.6) | 0.77 (0.6 to 0.99) | −2.8 (−5.6 to 0.0) | 0.77 (0.6 to 0.98) | −2.9 (−5.6 to −0.1) |
| 5th (wealthiest) | 7.0 (5.3 to 8.6) | 0.57 (0.42 to 0.77) | −5.3 (−8.4 to −2.3) | 0.56 (0.42 to 0.76) | −5.4 (−8.4 to −2.3) |
| **Cluster level** | Across quintiles | | | | |
| Wealth index quintile continuous | 9.9 (8.5 to 11.2) | 0.85 (0.77 to 0.94) | −1.6 (−2.6 to −0.5) | 0.85 (0.77 to 0.94) | −1.5 (−2.6 to −0.5) |
| Wealth Index quintile | In each quintile | | | | |
| 1st (least wealthy) | 13.1 (9.1 to 17.2) | ref | ref | ref | ref |
| 2nd | 10.6 (7.2 to 14.1) | 0.81 (0.52 to 1.27) | −2.5 (−7.8 to 2.8) | 0.85 (0.54 to 1.32) | −2.0 (−7.2 to 3.3) |
| 3rd | 9.7 (7.3 to 12.0) | 0.73 (0.5 to 1.09) | −3.5 (−8.2 to 1.2) | 0.77 (0.52 to 1.15) | −2.9 (−7.6 to 1.7) |
| 4th | 8.9 (5.8 to 12.0) | 0.68 (0.43 to 1.07) | −4.2 (−9.3 to 0.8) | 0.69 (0.44 to 1.09) | −4.0 (−8.9 to 1.0) |
| 5th (wealthiest) | 6.3 (4.2 to 8.5) | 0.48 (0.31 to 0.76) | −6.8 (−11.3 to −2.2) | 0.49 (0.31 to 0.77) | −6.5 (−10.9 to −2.1) |

Note: IRR-Incidence Rate Ratio, IRD-Incidence Rate Difference, P = 0.0004 and P = 0.0018 for test of linear trend in categories for household and cluster level respectively.

**Table 3. Estimates of inequality indices: relative index, slope index, and concentration index at household and cluster level.**

| | Point Estimate (95%CI) |
| --- | --- |
| **Household level** | |
| Relative Index of Inequality (RII) | 1.12 (1.05 to 1.19), P = 0.00 |
| Slope Index of Inequality (SII): average difference in mortality per 1000 PY | 1.50 (0.5 to 2.5), P = 0.004 |
| Gini Index | 0.59 (0.56 to 0.62), P = 0.000 |
| Concentration Index | −0.14 (−0.25 to −0.04), P = 0.007 |
| **Cluster level** | |
| Relative Index of Inequality (RII) | 1.17 (1.05 to 1.29), P = 0.00 |
| Slope Index of Inequality (SII): average difference in mortality per 1000 PY | 2.30 (0.2 to 4.4), P = 0.034 |
| Gini Index | 0.40 (0.37 to 0.43), P = 0.00 |
| Concentration index | −0.15 (−0.24 to −0.06), P = 0.002 |

Note: Estimates with P < 0.05 were considered statistically significant at α = 0.05. P-values are presented for reference.

the change in mortality rate by wealth was slightly more pronounced at cluster level, with a 15% reduction for each increase in quintile level (IRR = 0.85, 95% CI: 0.77 to 0.94).

The wealth related inequalities in mortality were also slightly larger at the community level with RII of 1.17, 95%CI (1.05 to 1.29) and SII: 2.3 per 1000-person year, 95%CI (0.2 to 4.4). These disparities, though small, were statistically significant. The observed changes in mortality by wealth quintiles remained consistent when adjusting for distance to the facility (Table 2).

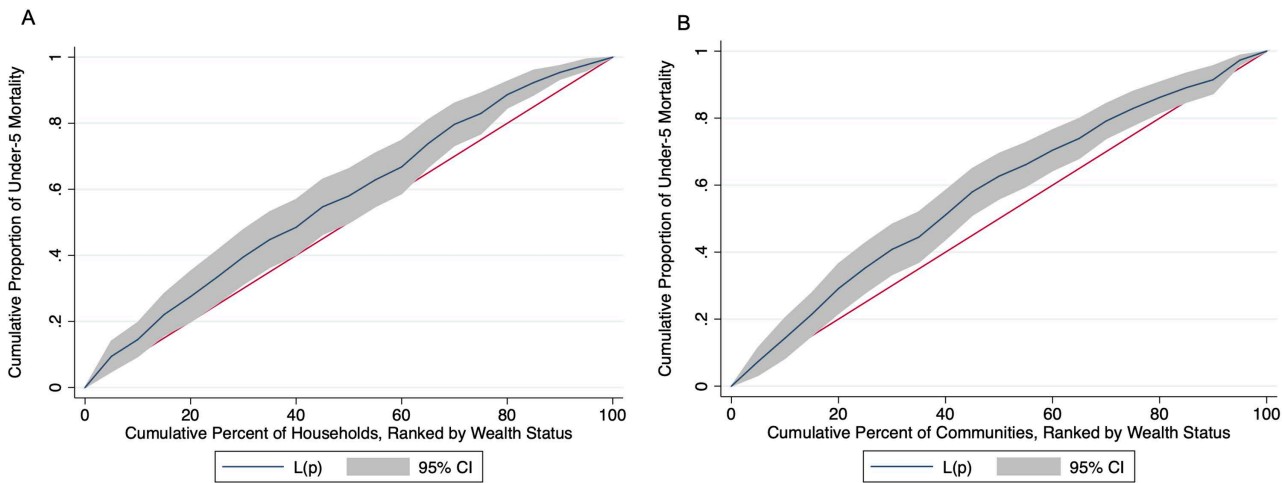

**Note:** L(P)- Cumulative Proportion of Under-5 Mortality at Wealth Percentile p. A concentration curve above the 45° line indicates inequality disfavoring the poor and corresponds to negative concentration index values.

**Fig 2. Concentration index of child mortality by wealth status at household and community levels.**

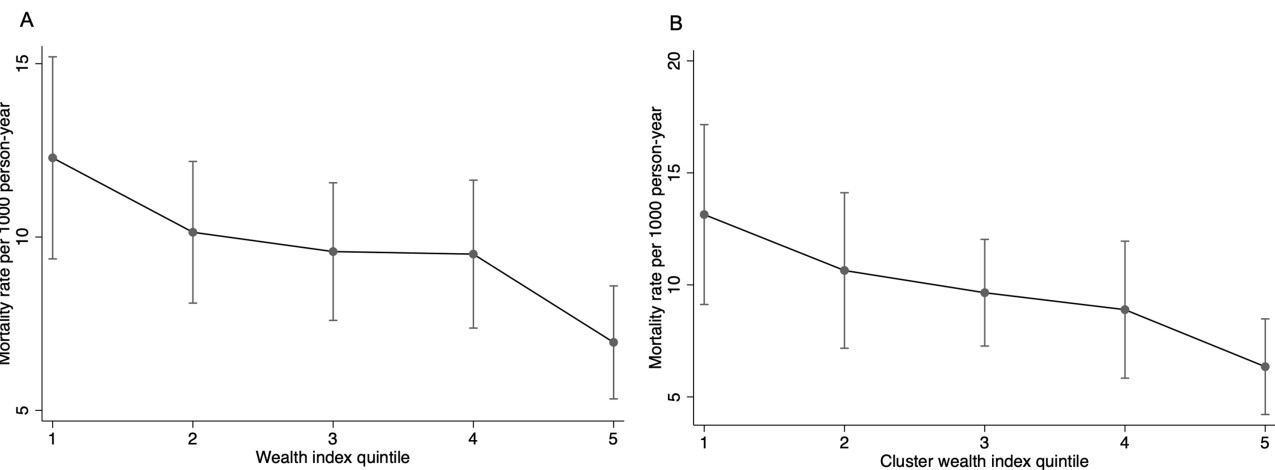

**Note:** Wealth index quintile 1 represents the lowest 20% of the population by wealth, while quintile 5 represents the highest 20%

**Fig 3. Mortality rate by a) household wealth index quintile and b) wealth at cluster level.**

## Wealth-AZ interaction

Mortality rates decreased with increasing wealth quintiles for both AZ and placebo clusters (IRR for AZ = 0.87, 95% CI: 0.80 to 0.95; IRR for placebo = 0.90, 95% CI: 0.82 to 0.99). Across all quintiles, the mortality rate was lower in the AZ group compared to the placebo group (IRR = 0.83, 95% CI: 0.63 to 1.1). The effect of AZ compared to placebo did not change significantly with the wealth index, and the change in mortality rate by wealth did not vary significantly between the arms (Table 4, Fig 4). Although there was a slight difference in the pattern of change at the household versus cluster level (Fig 4), there was no significant interaction between AZ and wealth on either the multiplicative or additive scale at both the household and cluster levels (Table 4). These results were consistent in the sensitivity analyses using a mixed-effects

Poisson model with a random intercept for cluster. The multiplicative interaction term remained non-significant (Interaction coefficient = 0.97, 95% CI: 0.86 to 1.09, *p* = 0.63), as did the additive interaction estimate (RERI = −0.03, 95% CI: −0.19 to 0.13, *p* = 0.72). Differences in RII and SII by arm were also minimal (Table 5).

### Sensitivity analyses

In the sensitivity analysis using the first 4 components, the results were similar (S1 and S2 Figs). We found that that the probability of households being included in the pre-census (selected) did not vary significantly by study arm (RR = 1.14, 95%CI (0.9 to 1.44). Furthermore, the effect of AZ on mortality did not vary substantially between households included in pre-census (selected) and missing households (P value for AZ*selection interaction = 0.862, S1-S2 Table).

### Discussion

We found a wealth gradient in child mortality at both household and community levels, with more pronounced disparities at the community level. At both levels, those in the least wealthy quintile had the highest mortality rates. These wealth-related disparities were similar in both AZ and placebo groups. Although AZ-treated groups consistently had lower mortality rates, the effect of AZ compared to placebo did not vary significantly across wealth quintiles. There was no evidence of a significant interaction between AZ and wealth status on either a multiplicative or additive scale at the household or cluster levels.

**Table 4. Interaction between wealth and treatment on under-5 mortality at household and cluster level.**

| Household level | Azithromycin | Placebo | IRR (AZ vs placebo) | IRD (AZ vs placebo) per 1000PY |
|---|---|---|---|---|
| Mortality rate per 1000 PY | | | | |
| Across quintiles | 8.9 (7.3 to 10.5) | 10.7 (8.3 to 13.1) | 0.83 (0.63 to 1.1) | −1.8 (−4.7 to 1.1) |
| By wealth index quintile | | | | |
| 1st | 11.5 (8.6 to 14.3) | 13.0 (8.8 to 17.1) | 0.88 (0.53 to 1.24) | −1.5 (−6.5 to 3.5) |
| 2nd | 10.0 (8.0 to 12.0) | 11.7 (8.7 to 14.7) | 0.86 (0.58 to 1.13) | −1.7 (−5.3 to 1.9) |
| 3rd | 8.7 (7.2 to 10.2) | 10.6 (8.2 to 12.9) | 0.83 (0.59 to 1.06) | −1.8 (−4.6 to 1.0) |
| 4th | 7.6 (6.2 to 9.1) | 9.5 (7.4 to 11.7) | 0.80 (0.56 to 1.04) | −1.9 (−4.5 to 0.7) |
| 5th | 6.7 (5.1 to 8.2) | 8.6 (6.3 to 10.9) | 0.77 (0.5 to 1.05) | −2.0 (−4.7 to 0.8) |
| Interaction Coeff multiplicative scale 0.97 (0.85 to 1.1), P = 0.607 | | | | |
| Interaction Coeff additive scale (RERI) −0.02 (−0.19 to 0.15), P = 0.823 | | | | |
| **Cluster level** | | | | |
| Mortality rate per 1000 PY | | | | |
| Across quintiles | 8.9 (7.3 to 10.5) | 10.7 (8.3 to 13.1) | 0.83 (0.63 to 1.1) | −1.8 (−4.7 to 1.1) |
| By wealth index quintile | | | | |
| 1st | 11.9 (8.2 to 15.6) | 14.9 (9.8 to 20) | 0.80 (0.43 to 1.17) | −3.0 (−9.3 to 3.3) |
| 2nd | 10.0 (7.8 to 12.1) | 12.7 (9.5 to 15.8) | 0.79 (0.53 to 1.05) | −2.7 (−6.5 to 1.1) |
| 3rd | 8.4 (7 to 9.7) | 10.8 (8.4 to 13.1) | 0.78 (0.57 to 0.99) | −2.4 (−5.1 to 0.3) |
| 4th | 7.0 (5.7 to 8.3) | 9.2 (6.7 to 11.6) | 0.77 (0.51 to 1.02) | −2.1 (−5.0 to 0.7) |
| 5th | 5.9 (4.2 to 7.5) | 7.8 (4.9 to 10.7) | 0.75 (0.4 to 1.11) | −1.9 (−5.3 to 1.4) |
| Interaction Coeff multiplicative scale 0.99 (0.82 to 1.19), P = 0.881 | | | | |
| Interaction Coeff additive scale (RERI) 0.02 (−0.23 to 0.27), P = 0.884 | | | | |

**Note:** IRR-Incidence Rate Ratio, IRD-Incidence Rate Difference. Wealth quintiles were used as linear form in estimating effect of wealth on mortality by treatment arm. Distance to facility was included as covariate. Slope Index of Inequality (SII) showing average difference in mortality per 1000 PY.

In the household-level sensitivity analysis using a mixed-effects Poisson model with a random intercept for cluster, the interaction coefficients were; multiplicative = 0.97 (95% CI: 0.86 to 1.09, p = 0.632); additive (RERI) = −0.03 (95% CI: −0.19 to 0.13, p = 0.724).

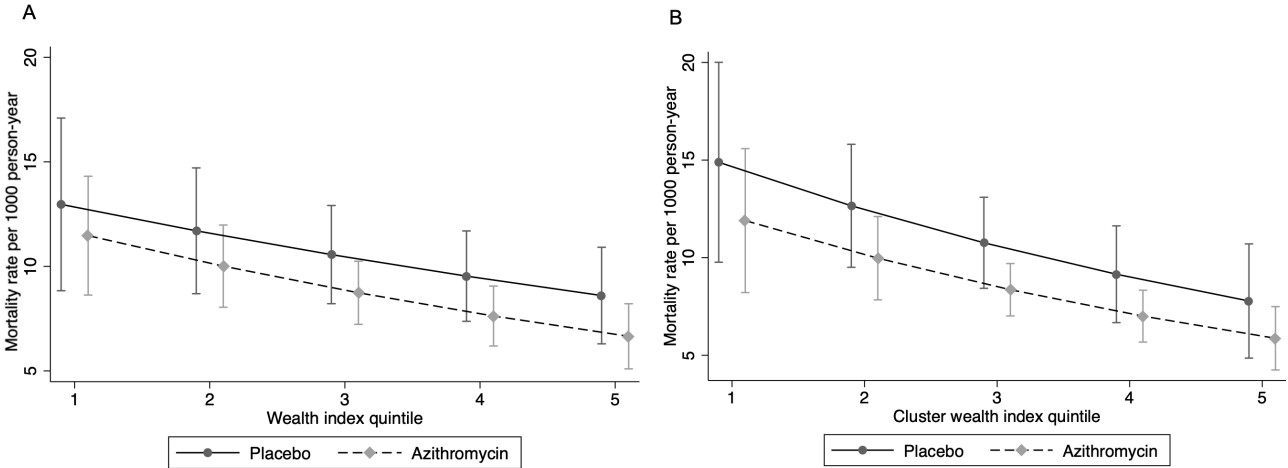

**Note:** models included distance to facility as covariate

**Fig 4. Mortality rate by treatment and wealth index quintile at a) household level and b) cluster level.**

**Table 5. Estimates of inequality indices by treatment arm at household and cluster level.**

| Household level | Azithromycin | Placebo |
|---|---|---|
| IRR (wealth->mortality) | 0.87 (0.8 to 0.95) | 0.90 (0.82 to 0.99) |
| IRD (wealth->mortality) per 1000 PY | −1.20 (−2.0 to −0.4) | −1.10 (−2.2 to 0.0) |
| Relative Index of Inequality (RII) | 1.15 (1.05 to 1.24), P=0.00 | 1.11 (1.01 to 1.2), P=0.00 |
| Slope Index of Inequality (SII) | 1.70 (0.30 to 3.10), P=0.019 | 1.40 (−0.10 to 2.90), P=0.072 |
| Gini Index | 0.59 (0.54 to 0.63), P=0.00 | 0.58 (0.54 to 0.62), P=0.00 |
| Concentration index | −0.20 (−0.32 to −0.08), P=0.002 | −0.09 (−0.26 to 0.08), P=0.289 |
| **Cluster level** | | |
| IRR (wealth->mortality) | 0.84 (0.74 to 0.95) | 0.85 (0.74 to 0.98) |
| IRD (wealth->mortality) per 1000 PY | −1.50 (−2.70 to −0.3) | −1.80 (−3.5 to −0.1) |
| Relative Index of Inequality (RII) | 1.19 (1.04 to 1.34), P=0.00 | 1.18 (1.0 to 1.35), P=0.00 |
| Slope Index of Inequality (SII) | 2.30 (−0.30 to 4.9), P=0.084 | 2.60 (−0.8 to 6.1), P=0.134 |
| Gini Index | 0.41 (0.37 to 0.46), P=0.00 | 0.38 (0.34 to 0.41), P=0.00 |
| Concentration index | −0.17 (−0.31 to −0.03), P=0.021 | −0.13 (−0.25 to −0.01), P=0.032 |

## Wealth and mortality

We found that mortality rates declined substantially with increasing wealth, consistent with previous studies that identified wealth as a key factor in under-5 mortality variability [4,43]. Wealth may influence child mortality in several ways. At the community level, factors such as ecological settings (e.g., local environment), political economy (e.g., economic conditions and governance), and health systems (e.g., access to healthcare services) can all influence child mortality [22]. At the household level, factors linked to a family's income and wealth can influence access to goods and services—such as housing, food, transportation, and healthcare [22,44]. These factors can individually and collectively affect mortality through more proximal factors like poorer living conditions, higher susceptibility to infections, and reduced access to quality care [22,45,46], likely contributing to the higher child mortality observed among the poorest households and communities. Additional barriers include limited access to routine services, such as timely childhood vaccination, which are often

lower among the most disadvantaged populations in Sub-Saharan Africa [47]. Distance to facilities and the inability to afford care—factors commonly associated with poverty—have been noted as reasons for such disparities and as general barriers to care [15,48,49]. However, in our study, adjusting for distance to facilities did not change the disparities in mortality observed by wealth.

Over the years, Burkina Faso has made notable progress in reducing childhood mortality by increasing access to free care, adopting seasonal malaria chemoprevention, expanding services, and improving the quality of care [50]. While free healthcare for children under 5 has improved healthcare utilization and helped reduce inequalities [51], some disparities remain, with the least wealthy households and communities experiencing the highest mortality rates. The modest change in mortality rates for those in the 40th to 80th percentiles suggests that the wealth impacts on under 5 mortality may be most pronounced in the wealthiest and poorest 20%. A closer examination of how wealth translates to health outcomes at the extremes of the wealth spectrum could be helpful. Although the Gini index showed higher variability in child mortality across households than communities, disparities by wealth appeared to be higher at the community level, as seen with the relative and slope inequality indices, as well as the concentration index. The more pronounced community-level disparities (than the household level) may suggest that aggregate factors—such as healthcare accessibility, environmental conditions, and community resources—may play a larger role, especially in the context of residential segregation by wealth. Targeted approaches addressing the specific challenges of vulnerable populations [52], including enhancing community resilience, [53] may further reduce differential mortality.

Although the analysis may have been underpowered to detect modest interaction effects, we did not find evidence of an interaction between AZ and wealth on mortality rates. The wealth gradient in child mortality was similar in both AZ and placebo-treated clusters. Previous research indicated that AZ could help reduce health disparities by being particularly beneficial for distant communities with reduced access to healthcare and antibiotics. [13] Although the mechanism of action is not clear, AZ may improve short-term health outcomes by clearing infections [24], which is critical given that many infectious diseases can worsen rapidly without treatment. This can make AZ particularly effective at reducing mortality and morbidity in populations that are geographically disadvantaged, where treatment access is limited. While distance to facilities is a significant barrier to accessing routine and life-saving interventions [14,15], it often represents one of many obstacles to care. Thus, while mass AZ distribution may also help address some of the wealth-related disparities due to barriers to treatment and care, its effectiveness in addressing broader disparities influenced by factors such as economic inequality, nutrition, and living conditions may be more limited.

Regardless of its impact on wealth-related disparities, AZ remains effective in reducing child mortality [8,9,10], as reflected with the lower mortality rates in AZ treated communities. Therefore, using AZ with comprehensive approaches that strengthen health systems and services in poor communities, improve living conditions and nutritional status of disadvantaged children, increase awareness and coverage of interventions, and enhance the quality and equity of care may better help address wealth-related disparities [52].

The consistent effect of AZ across different wealth levels may imply that it provides similar

benefits across economically diverse communities, with no evidence suggesting enhanced benefits for disadvantaged communities or for prioritizing treatment based on wealth status. Previous studies exploring the heterogeneity of AZ's effect by underlying mortality rate [54] or by nutritional status [55] did not find strong evidence of effect modification. While concerns about resistance call for targeted approaches [56], AZ appears beneficial for a broad range of populations. Findings from the AVENIR trial also showed that nontargeted intervention for all children aged 1–59 months yielded a greater benefit against mortality compared to restricting treatment to infants 1–11 months [10]. While it is important to explore whether certain populations benefit more from MDA with AZ and prioritize them accordingly, continuing to treat all children who can benefit from it remains crucial in the absence of evidence for a targeted approach or better alternatives [57]. A study weighing the risks and benefits of MDA AZ found that the short term benefits outweigh the risks, as AZ

is an effective short-term strategy for reducing mortality that can be used along with larger-scale structural and systemic changes [58].

This study has some limitations. The first is reduced statistical power due to relatively low mortality rates. The need to restrict the analysis to households included in the pre-census, i.e., the exclusion of missing households—further lowered our sample size. Analyzing data at the cluster level partly mitigated this issue by averaging household wealth scores and aggregating mortality by cluster, allowing us to incorporate mortality information from households missing wealth scores. This, however, relied on the assumption that the households with non-missing scores can adequately represent those with missing scores. This approach assumes homogeneity within clusters, potentially masking intra-cluster variability in wealth. Additionally, our sensitivity analyses suggest that the missingness of households in the pre-census appears to be a random process and does not differ systematically by study arm. The scenario shown in Fig 1D indicates that selection can be a threat to generalizability of findings to target population if S (selection) for inclusion in the analysis is associated with changes in mortality by wealth ($\triangle Y_W$) [41]. Our sensitivity analysis shows that treatment (A) is not associated with selection (S), thereby breaking the link between these variables. Additionally, the effect of AZ on mortality in the selected and missing households was also similar. This suggests that the threat to validity shown in Fig 1C may not apply, since there is no association between selection into the sample (S) (selection) and the effect of AZ on mortality ($\triangle Y_A$) in our data. This implies that the observed effects in the selected population can be generalized to the full sample and target population [41].

Second, there is a potential for misclassification or measurement error, as the asset-based wealth index may not fully capture the complexity and multidimensional nature of wealth within households. Although this limitation could affect the accuracy of our analyses, it would likely impact all communities similarly and would not be differential measurement error by mortality levels. The extent to which these scores adequately represent wealth can also be affected by the PCA. Although the proportion of variability explained by the first component was not large and may therefore be an oversimplification, the combination of the first 4 components, which explained a much larger proportion of variability, generated similar findings. Additionally, a summary of household characteristics by wealth index quintile shows that asset ownership increases with wealth, as expected. This trend further strengthens our confidence in using the scores generated from PCA to represent wealth status (S1 Table). Lastly, since we did not have additional data on clusters and households, we could not adjust for other potential confounders of the wealth-mortality association, such as bed net access, nutritional status, immunization coverage, or environmental exposures. Therefore, the objective was to examine how mortality rates vary with wealth in these communities and examine how it interacts with AZ, rather than to estimate the causal effect of asset-based wealth on child mortality.

## Conclusion

Our results suggest that while wealth-related disparities in child mortality were observed at both the household and community levels, mass AZ treatment did not appear to mitigate these disparities, highlighting the need for other targeted approaches that could address this differential mortality. We did not find evidence of an enhanced benefit of AZ for disadvantaged communities or for prioritizing treatment based on wealth status. Therefore, it may be beneficial for MDA AZ programs to continue providing treatment to all children who could benefit. Future work is needed to identify and evaluate targeted strategies that address the underlying drivers of disparities in child mortality across wealth levels.

## Supporting information

**S1 Table. Sensitivity Analyses Assessing the Role of Selection.**
(TIFF)

**S2 Table. Summary of household characteristic by wealth index quintiles.**
(TIFF)

**S1 Fig. Mortality rate by a) Household wealth index quintile and b) wealth at cluster level with wealth score generated from 4 principal components.**
(TIFF)

**S2 Fig. Mortality rate by treatment and wealth index quintile at a) household level and b) cluster level with wealth score generated from 4 principal components.**
(TIFF)

## Author contributions

**Conceptualization:** Elisabeth A. Gebreegziabher, Ali Sié, Michelle S. Hsiang, David V. Glidden, Benjamin F. Arnold, Thomas M. Lietman, Catherine E. Oldenburg.

**Data curation:** Valentin Boudo, Huiyu Hu.

**Formal analysis:** Elisabeth A. Gebreegziabher.

**Funding acquisition:** Ali Sié, Thomas M. Lietman, Catherine E. Oldenburg.

**Investigation:** Elisabeth A. Gebreegziabher.

**Methodology:** Elisabeth A. Gebreegziabher, Pearl Anne Ante-Testard, Steven E. Gregorich, Michelle S. Hsiang, David V. Glidden, Benjamin F. Arnold.

**Project administration:** Ali Sié, Mamadou Ouattara, Mamadou Bountogo, Boubacar Coulibaly, Thierry Ouedraogo, Elodie Lebas, Thomas M. Lietman, Catherine E. Oldenburg.

**Resources:** Mamadou Ouattara, Mamadou Bountogo, Boubacar Coulibaly, Thierry Ouedraogo, Elodie Lebas.

**Software:** Elisabeth A. Gebreegziabher.

**Supervision:** Ali Sié, Mamadou Ouattara, Mamadou Bountogo, Boubacar Coulibaly, Thierry Ouedraogo, Elodie Lebas, Thomas M. Lietman, Catherine E. Oldenburg.

**Visualization:** Elisabeth A. Gebreegziabher.

**Writing – original draft:** Elisabeth A. Gebreegziabher.

**Writing – review & editing:** Elisabeth A. Gebreegziabher, Ali Sié, Mamadou Ouattara, Mamadou Bountogo, Boubacar Coulibaly, Valentin Boudo, Thierry Ouedraogo, Elodie Lebas, Huiyu Hu, Pearl Anne Ante-Testard, Steven E. Gregorich, Kieran S. O'Brien, Michelle S. Hsiang, David V. Glidden, Benjamin F. Arnold, Thomas M. Lietman, Catherine E. Oldenburg.

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
