## [Decision Letter · Decision Letter 0]

11 Jul 2025

Dear Dr. Gebreegziabher,

Thank you for submitting your manuscript to PLOS ONE. After careful consideration, we feel that it has merit but does not fully meet PLOS ONE’s publication criteria as it currently stands. Therefore, we invite you to submit a revised version of the manuscript that addresses the points raised during the review process.

We look forward to receiving your revised manuscript.

Kind regards,

Saidul Abrar, MBBS, MPH, Ph.D

Academic Editor

PLOS ONE

**Journal Requirements:**

1. When submitting your revision, we need you to address these additional requirements. Please ensure that your manuscript meets PLOS ONE's style requirements, including those for file naming. The PLOS ONE style templates can be found at  https://journals.plos.org/plosone/s/file?id=wjVg/PLOSOne_formatting_sample_main_body.pdf and  https://journals.plos.org/plosone/s/file?id=ba62/PLOSOne_formatting_sample_title_authors_affiliations.pdf 2. Thank you for stating in your Funding Statement: The CHAT trial was supported by the Gates Foundation (grant number OPP1187628). Research reported in this manuscript was also supported by the National Institutes of Health Eunice Kennedy Shriver National Institute of Child Health & Human Development (NIH/NICHD) F31 Award (1F31HD114434-01A1: E.A.G.).  Please provide an amended statement that declares *all* the funding or sources of support (whether external or internal to your organization) received during this study, as detailed online in our guide for authors at http://journals.plos.org/plosone/s/submit-now.  Please also include the statement “There was no additional external funding received for this study.” in your updated Funding Statement. Please include your amended Funding Statement within your cover letter. We will change the online submission form on your behalf. 3. In the online submission form, you indicated that your data will be submitted to a repository upon acceptance.  We strongly recommend all authors deposit their data before acceptance, as the process can be lengthy and hold up publication timelines. Please note that, though access restrictions are acceptable now, your entire minimal dataset will need to be made freely accessible if your manuscript is accepted for publication. This policy applies to all data except where public deposition would breach compliance with the protocol approved by your research ethics board. If you are unable to adhere to our open data policy, please kindly revise your statement to explain your reasoning and we will seek the editor's input on an exemption. 4. Your ethics statement should only appear in the Methods section of your manuscript. If your ethics statement is written in any section besides the Methods, please move it to the Methods section and delete it from any other section. Please ensure that your ethics statement is included in your manuscript, as the ethics statement entered into the online submission form will not be published alongside your manuscript.

**Additional Editor Comments:**

Kindly revise your manuscript as per the comments of reviewers. Thanks

Reviewers' comments:

Reviewer's Responses to Questions

**Comments to the Author**

1. Is the manuscript technically sound, and do the data support the conclusions?

Reviewer #1: Partly

Reviewer #2: Yes

2. Has the statistical analysis been performed appropriately and rigorously?

Reviewer #1: No

Reviewer #2: Yes

3. Have the authors made all data underlying the findings in their manuscript fully available?

Reviewer #1: Yes

Reviewer #2: Yes

4. Is the manuscript presented in an intelligible fashion and written in standard English?

Reviewer #1: Yes

Reviewer #2: Yes

**Reviewer #1:**  As the statistical reviewer I will focus on methods and reporting.

Major

1) One major concern is where AZ exposure sits on the causal pathway in relation to SES and the primary outcome. I appreciate this was on the back of an RCT, so in principle everyone should be exposed. but can the authors explore and confirm that? Perhaps another short section on wealth status and AZ exposure in the methods section. In other words, the study’s observational secondary analysis design limits causal inference regarding the interaction between azithromycin and wealth, particularly in the absence of randomisation by wealth strata.

2) why isn't the village cluster included in the models as an additional higher level random intercept? a 3 level mixed effects model would be preferable to the current model (clarify which command in Stata you used and you will use btw) with robust standard errors.

3) Power will be pretty low for the interaction analysis, something the authors state in the limitations section, but they discuss statistical power in the abstract and the results. the abstract and the discussion needs to reflect that power probably too low to pick up anything but a very large association. so be more conservative when you discuss the findings from the interaction analysis.

4) The strategy for dealing with missing data needs to be clearly stated. There is a brief mention in the context of the sensitiivty analysis, but as per the STROBE statement (which you should follow) a clear explanation is needed about the approach (in this case listwise deletion). My issue is why multiple imputation was not conducted as a sensitivity analysis (or the main analysis preferably). This is a major concern since nearly half the households from the household-level analysis are excluded due to missing pre-census data, something that could introduce selection bias, even if sensitivity analyses suggest minimal impact.

6) PCI may oversimplify the multidimensional nature of socioeconomic status and introduce measurement error, especially given the relatively low variance explained by the first component (discuss as a limitation).

7) The use of community-level averages to impute wealth in cluster-level analyses assumes homogeneity within clusters, potentially masking intra-cluster variability.

Minor

1) clarify for readers that the random components are ignored with the margins command.

2) rephrase "robust standard errors clustered at the cluster level to account for the cluster-level treatment in..."

3) Another limitation to discuss is potential unmeasured confounders such as nutritional status, immunization coverage, or environmental exposures which may influence both wealth and mortality

**Reviewer #2:**  The authors have done tremendous work for the clinical trials, but still the manuscript needs to be revised for some of the suggestions prior to being published.

1) The abstract section must be revised, as a lot of changes are recommended in the file section.

2) The manuscript must be revised for typographic mistakes and grammatical errors as highlighted in the file section.

3) The cluster clinical trials are not explained. Are the trials single-blind or double-blind?

4) The result and discussion portion should be updated with strong, updated references.

5) The conclusion section should be written scientifically with sophisticated words.

6) There is no "future prospect" section included in the manuscript?

7) If the data showed in the manuscript is directly taken from the patients, then the manuscript is acceptable for publication, but if the analyzed data is taken from any already collected source, then the manuscript should not be published.

8) Which software was used to analyze the data?

**Do you want your identity to be public for this peer review?** For information about this choice, including consent withdrawal, please see our Privacy Policy

Reviewer #1: No

Reviewer #2: No

---

## [Author Response · Author response to Decision Letter 1]

26 Aug 2025

August 23, 2025

Academic Editor: Saidul Abrar, MBBS, MPH, Ph.D

PLOS One

RE: MANUSCRIPT ENTITLED, “Exploring Heterogeneity in Treatment Effects: The Impact and Interaction of Asset-Based Wealth and Mass Azithromycin Distribution on Child Mortality” PONE-D-25-23672

Dear Dr. Saidul Abrar,

We are grateful to have the opportunity to respond to the Reviewers’ and Editor’s comments and queries. We truly appreciate their time and effort in assessing our manuscript. Our responses follow below:

Journal Requirements:

Response: We have revised the manuscript to align with PLOS ONE’s formatting requirements, using the style templates provided.

The CHAT trial was supported by the Gates Foundation (grant number OPP1187628). Research reported in this manuscript was also supported by the National Institutes of Health Eunice Kennedy Shriver National Institute of Child Health & Human Development (NIH/NICHD) F31 Award (1F31HD114434-01A1: E.A.G.).

Response: As suggested, we have added in the funding statement that no additional external funding was received for this study.

3. In the online submission form, you indicated that your data will be submitted to a repository upon acceptance. We strongly recommend all authors deposit their data before acceptance, as the process can be lengthy and hold up publication timelines. Please note that, though access restrictions are acceptable now, your entire minimal dataset will need to be made freely accessible if your manuscript is accepted for publication. This policy applies to all data except where public deposition would breach compliance with the protocol approved by your research ethics board. If you are unable to adhere to our open data policy, please kindly revise your statement to explain your reasoning and we will seek the editor's input on an exemption.

Response: Data will be made available in OSF upon acceptance. We are currently finalizing the process to publish the data. We have noted this in the manuscript.

Response: We have ensured the ethics statement appears exclusively in the Methods section as required.

Additional Editor Comments:

Kindly revise your manuscript as per the comments of reviewers. Thanks

Reviewers' comments:

Reviewer's Responses to Questions

Comments to the Author

Reviewer #1: As the statistical reviewer I will focus on methods and reporting.

Major

1) One major concern is where AZ exposure sits on the causal pathway in relation to SES and the primary outcome. I appreciate this was on the back of an RCT, so in principle everyone should be exposed. but can the authors explore and confirm that? Perhaps another short section on wealth status and AZ exposure in the methods section. In other words, the study’s observational secondary analysis design limits causal inference regarding the interaction between azithromycin and wealth, particularly in the absence of randomisation by wealth strata.

Response: We appreciate the reviewer’s point about the limitations introduced by the lack of randomization of wealth. AZ was randomized at the community level, which implies that, by design, treatment assignment is independent of household or community wealth status, demographics, or other factors. This eliminates any influence on AZ exposure. Furthermore, treatment coverage (of assigned treatment, AZ or placebo) was approximately 90% in both arms, and all communities received their assigned treatment per protocol, ensuring no differential implementation across wealth strata. As suggested, to confirm the balance of treatment assignment across SES strata, we evaluated the proportion treated by wealth quintile. AZ and placebo were distributed approximately equally across all five wealth quintiles, with no systematic differences (e.g., 58.8% AZ in the poorest quintile vs. 50.0% in the wealthiest quintile). These percentages for each wealth quintile have been added to the Results section to clarify this point.

While we acknowledge the lack of randomization of wealth may limit causal interpretation of the interaction, our analysis helps us evaluate whether the causal effect of AZ estimated through randomization, varies across levels of wealth. This approach aligns with standard effect modification analyses in RCTs and provides policy-relevant insight into whether AZ offers differential benefits by socioeconomic status. We have also avoided language suggesting causal interaction, in line with this distinction.

We have added the following in the beginning of the results section. “The overall coverage of MDA was approximately 90% for both AZ and placebo arms. The proportion of communities that received AZ in each wealth quintile was: 58.8% (quintile 1, poorest), 47.1% (quintile 2), 58.8% (quintile 3), 50.0% (quintile 4), and 50.0% (quintile 5, wealthiest).”

2) why isn't the village cluster included in the models as an additional higher level random intercept? a 3 level mixed effects model would be preferable to the current model (clarify which command in Stata you used and you will use btw) with robust standard errors.

Response: We appreciate the suggestion regarding model choice. For the household-level interaction analysis, we used a Poisson regression model (poisson in Stata) with a log link, person-years as an offset, and robust standard errors clustered at the level of randomization (vce(cluster clusterid)) to account for correlation between households within the same cluster. Clusters and villages were used interchangeably throughout the manuscript. We have now updated it to clusters or communities. Our intent was to estimate how the effect of azithromycin varies by wealth at the population level (i.e., marginal effects), rather than to model variation between communities, as would be done in a mixed-effects framework.

We agree, however, that it is important to assess whether the results are sensitive to the choice of model. In response to this comment, we conducted a sensitivity analysis using a mixed-effects Poisson model (mepoisson) with a random intercept for cluster.

The results from this model were consistent with our main analysis. Specifically, the interaction between azithromycin and wealth remained non-significant on both the multiplicative and additive scales: Multiplicative interaction coefficient: 0.97 (95% CI: 0.86 to 1.09, p = 0.632); Additive interaction (RERI): –0.03 (95% CI: –0.19 to 0.13, p = 0.724)

This additional analysis and its findings are now reported in the corresponding Methods and Results sections and noted in a footnote to Table 4.

3) Power will be pretty low for the interaction analysis, something the authors state in the limitations section, but they discuss statistical power in the abstract and the results. the abstract and the discussion needs to reflect that power probably too low to pick up anything but a very large association. so be more conservative when you discuss the findings from the interaction analysis.

Response: We agree and have revised the abstract and discussion to more explicitly acknowledge the limited power for detecting interaction effects. We now note that the analyses may have been underpowered to detect small to moderate effects. We have added the phrase: “While the analysis may have been underpowered to detect modest interaction effects, …” in both sections.

4) The strategy for dealing with missing data needs to be clearly stated. There is a brief mention in the context of the sensitivity analysis, but as per the STROBE statement (which you should follow) a clear explanation is needed about the approach (in this case listwise deletion). My issue is why multiple imputation was not conducted as a sensitivity analysis (or the main analysis preferably). This is a major concern since nearly half the households from the household-level analysis are excluded due to missing pre-census data, something that could introduce selection bias, even if sensitivity analyses suggest minimal impact.

Response: As suggested, we have clarified in the Methods section that households with missing pre-census data were excluded from the household-level analysis using listwise deletion (i.e., complete case analysis) to explicitly state this approach and align with the STROBE guidelines on missing data reporting.

Multiple imputation was considered but was not feasible in this case. The pre-census data had nearly all household-level variables, which were missing for these excluded households, leaving only mortality, treatment assignment (AZ versus placebo), follow-up time, and community ID documented in the main trial. These variables were insufficient to reliably impute missing wealth scores.

In addition to conducting parallel analyses at the community level, we assessed whether the missingness process may have introduced selection bias. Sensitivity analyses suggested that missingness was not associated with study arm and that the effect of AZ on mortality was similar among included and excluded households. This provides some reassurance that the exclusion due to missing data did not substantially bias the results. We have also acknowledged this issue and our sensitivity findings in the Limitations section.

6) PCI may oversimplify the multidimensional nature of socioeconomic status and introduce measurement error, especially given the relatively low variance explained by the first component (discuss as a limitation).

Response: As suggested, we have highlighted the potential oversimplification of the multidimensional nature of wealth. The limitations paragraph has been revised to explicitly acknowledge that the PCA may not fully capture the complexity of household socioeconomic status and could introduce measurement error.

“Second, there is a potential for misclassification or measurement error, as the asset-based wealth index may not fully capture the complexity and multidimensional nature of wealth within households. Although this limitation could affect the accuracy of our analyses, it would likely impact all communities similarly and would not be differential measurement error by mortality levels. The extent to which these scores adequately represent wealth can also be affected by the PCA. Although the proportion of variability explained by the first component was not large and may therefore be an oversimplification, the combination of the first 4 components, which explained a much larger proportion of variability, generated similar findings.”

7) The use of community-level averages to impute wealth in cluster-level analyses assumes homogeneity within clusters, potentially masking intra-cluster variability.

Response: As suggested, we have acknowledged this limitation and added the following statement: “This approach assumes homogeneity within clusters, potentially masking intra-cluster variability in wealth.”

Minor

1) clarify for readers that the random components are ignored with the margins command.

Response: Thank you for this comment. We used the margins command only with the Poisson regression model that included clustered robust standard errors but no random effects. Therefore, no random components were ignored in the estimation. We have clarified in the Methods section that these estimates reflect marginal effects. For the mixed-effects Poisson model used in the sensitivity analysis, we directly reported the interaction coefficients on both the multiplicative and additive scales.

2) rephrase "robust standard errors clustered at the cluster level to account for the cluster-level treatment in..."

Response: As suggested, we have rephrased the text to: “robust standard errors clustered at the level of randomization (i.e., community/cluster) to account for cluster-level treatment in the household-level analysis.”

3) Another limitation to discuss is potential unmeasured confounders such as nutritional status, immunization coverage, or environmental exposures which may influence both wealth and mortality

Response: Thank you for the suggestion. We have updated the limitations section to acknowledge additional potential unmeasured confounders. The revised text now reads:

“Lastly, since we did not have additional data on clusters and households, we could not adjust for other potential confounders of the wealth–mortality association, such as bed net access, nutritional status, immunization coverage, or environmental exposures.”

Reviewer #2: The authors have done tremendous work for the clinical trials, but still the manuscript needs to be revised for some of the suggestions prior to being published.

1) The abstract section must be revised, as a lot of changes are recommended in the file section.

Response: As suggested, we have revised the abstract for clarity, scientific tone, and grammar, and incorporated changes in the sections highlighted in the file.

2) The manuscript must be revised for typographic mistakes and grammatical errors as highlighted in the file section.

Response: Thank you for your comment. We have reviewed the manuscript for typographic and grammatical errors and revised the text for clarity, grammar, and flow throughout, including in the highlighted sections.

3) The cluster clinical trials are not explained. Are the trials single-blind or double-blind?

Response: We have added in the Methods section, under Data Collection, that the CHAT trial was double-masked and placebo-controlled.

4) The result and discussion portion should be updated with strong, updated references.

Response: As suggested, we have replaced older citations, such as Peters (2008) and Liu (2012), with more recent and relevant publications. We have also reviewed all references to ensure they are up-to-date and appropriately support the results and discussion.

5) The conclusion section should be written scientifically with sophisticated words.

Response: We have revised the conclusion using a more scientific and polished tone, while maintaining the clarity and key messages of the original text.

6) There is no "future prospect" section included in the manuscript?

Response: We have made the implications for future research and programmatic focus, which are outlined in the concluding paragraph, more explicit. Specifically, the study highlights that despite reductions in overall child mortality, significant wealth-related disparities persist. We found no evidence to support prioritization of AZ distribution based on wealth status. Therefore, our findings support continued broad-based mass drug administration programs while also underscoring the need for further research int

---

## [Decision Letter · Decision Letter 1]

21 Nov 2025

Dear Dr. Gebreegziabher,

We look forward to receiving your revised manuscript.

Kind regards,

Alejandro Torrado Pacheco, PhD

Staff Editor

PLOS ONE

Journal Requirements:

**Additional Editor Comments:**

The manuscript has been assessed by two reviewers and their comments are available below. The reviewers request further clarification on the study design and statistics. Could you please carefully address all of the comments?

Reviewers' comments:

Reviewer's Responses to Questions

**Comments to the Author**

Reviewer #1: All comments have been addressed

Reviewer #3: (No Response)

2. Is the manuscript technically sound, and do the data support the conclusions?

Reviewer #1: Yes

Reviewer #3: Partly

3. Has the statistical analysis been performed appropriately and rigorously?

Reviewer #1: Yes

Reviewer #3: N/A

4. Have the authors made all data underlying the findings in their manuscript fully available?

Reviewer #1: Yes

Reviewer #3: No

5. Is the manuscript presented in an intelligible fashion and written in standard English?

Reviewer #1: Yes

Reviewer #3: Yes

Reviewer #1: I am satisfied with the authors' responses and the resulting changes to the paper. I have nothing else to add.

Reviewer #3: This is an interesting study examining the effect and interaction of asset-based wealth and mass azithromycin (AZ) distribution on child mortality. On first reading, I found the indication for AZ unclear, even with my background in pharmacy. I had to refer to the CHAT trial to infer the context of this study. The authors should also describe the indication for AZ treatment in the CHAT trial (e.g., seasonal chemoprevention of malaria) within the manuscript for better context and accuracy.

The authors used the mass distribution of AZ as an example of an intervention aimed at reducing child mortality, suggesting that AZ could buffer against disparities in mortality by addressing treatment gaps. However, the use of the term “AZ distribution for prevention of mortality” (line 120) may be too strong, as AZ does not have such an indication. A more accurate phrasing might be “prevention of mortality related to infection.”

The authors should justify why AZ, a macrolide antibiotic, was chosen when other antibiotics (e.g., broad-spectrum antibiotics) — potentially cheaper alternatives — could also be used to treat childhood infections. Furthermore, if AZ is being considered in relation to malaria, clarification is needed on why an antibiotic was selected over an antimalarial agent.

Line 193: Please specify the variables included in the Poisson regression model. Additionally, provide justification for using the principal component analysis method instead of factor analysis.

Minor:

Results — Table 3: Emphasize or clearly mark the statistically significant values.

**Do you want your identity to be public for this peer review?** For information about this choice, including consent withdrawal, please see our Privacy Policy

Reviewer #1: No

Reviewer #3: **Yes:** Ee Vien Low

---

## [Author Response · Author response to Decision Letter 2]

12 Dec 2025

December 12, 2025

Staff Editor: Alejandro Torrado Pacheco, PhD

PLOS One

RE: MANUSCRIPT ENTITLED, “Exploring Heterogeneity in Treatment Effects: The Impact and Interaction of Asset-Based Wealth and Mass Azithromycin Distribution on Child Mortality” PONE-D-25-23672

Dear Dr. Alejandro Torrado Pacheco,

We are grateful to have the opportunity to respond to the Reviewers’ and Editor’s comments and queries. We truly appreciate their time and effort in assessing our manuscript. Our responses follow below:

Review Comments to the Author

Reviewer #1: I am satisfied with the authors' responses and the resulting changes to the paper. I have nothing else to add.

Response: Thank you! We appreciate the reviewer’s time and helpful feedback.

Reviewer #3: This is an interesting study examining the effect and interaction of asset-based wealth and mass azithromycin (AZ) distribution on child mortality.

1. On first reading, I found the indication for AZ unclear, even with my background in pharmacy. I had to refer to the CHAT trial to infer the context of this study. The authors should also describe the indication for AZ treatment in the CHAT trial (e.g., seasonal chemoprevention of malaria) within the manuscript for better context and accuracy.

Response: We have clarified the indication and rationale for azithromycin use in the CHAT trial to provide appropriate clinical context. We now note that azithromycin was distributed biannually to children as part of a community-based intervention aimed at reducing childhood mortality. Although azithromycin is primarily used to treat bacterial infections, its potential mortality benefit in mass drug administration programs has been attributed to its broad antimicrobial activity against respiratory and enteric pathogens common in low-resource settings.

We have added the following in the methods line 134:

“In the CHAT trial, azithromycin was distributed biannually to children as part of a community-based intervention aimed at reducing childhood mortality. Although azithromycin is an antibiotic primarily used to treat bacterial infections, its potential mortality benefit in mass drug administration programs has been attributed to its broad antimicrobial activity, including against respiratory and enteric pathogens common in low-resource settings.”

2. The authors used the mass distribution of AZ as an example of an intervention aimed at reducing child mortality, suggesting that AZ could buffer against disparities in mortality by addressing treatment gaps. However, the use of the term “AZ distribution for prevention of mortality” (line 120) may be too strong, as AZ does not have such an indication. A more accurate phrasing might be “prevention of mortality related to infection.”

Response: As suggested, we have replaced the phrase “for prevention of mortality” with “for prevention of infection-related mortality” throughout the manuscript (introduction, line 77 and 87, and Methods line 134). This more accurately reflects the intended mechanism.

3. The authors should justify why AZ, a macrolide antibiotic, was chosen when other antibiotics (e.g., broad-spectrum antibiotics) — potentially cheaper alternatives — could also be used to treat childhood infections. Furthermore, if AZ is being considered in relation to malaria, clarification is needed on why an antibiotic was selected over an antimalarial agent.

Response: We appreciate the reviewer’s comment. We have added justification for the use of azithromycin in the Introduction. Specifically, we clarify that azithromycin was selected in the CHAT trial based on strong prior evidence from large randomized controlled trials—most notably the MORDOR trial—which demonstrated reductions in all-cause childhood mortality following community-level mass distribution of azithromycin in sub-Saharan Africa. These findings motivated subsequent trials, including CHAT, to further evaluate its population-level impact. We also highlight azithromycin’s favorable pharmacologic and safety profile, including its long half-life, broad antimicrobial spectrum (covering respiratory and enteric pathogens), and demonstrated feasibility for use in mass drug administration programs. In our study setting, malaria prevention was addressed through routine seasonal malaria chemoprevention (SMC) programs, and azithromycin was not intended to replace antimalarial agents but rather to act as a complementary intervention targeting other major infectious causes of childhood mortality such as pneumonia and diarrhea.

We have added the following text in the Introduction (line 77):

AZ was selected for mass-distribution trials based on strong prior evidence from large randomized controlled trials, including the MORDOR study in Malawi, Niger, and Tanzania, which demonstrated significant reductions in all-cause childhood mortality following community-wide AZ administration.[8] These results led to further investigations, including the CHAT trial in Burkina Faso and the AVENIR trial in Niger, to evaluate whether these benefits could be replicated in other populations and contexts.[9, 10, 11] The favorable pharmacologic properties of AZ including its long half-life, broad antimicrobial activity against respiratory and enteric pathogens, and well-established safety profile, make it suitable for large-scale community interventions.[12]

4. Line 193: Please specify the variables included in the Poisson regression model. Additionally, provide justification for using the principal component analysis method instead of factor analysis.

Response: As suggested, we have clarified the model specification and the rationale for using principal component analysis (PCA). In the Statistical Analyses Methods section, line 215, we added:

“… Poisson regression model for the wealth - mortality analyses included child mortality as the outcome, the household wealth index as the main exposure, person-years at risk as an offset, and robust standard errors clustered at the community level to account for the cluster-level treatment.” As noted in the manuscript, distance to the nearest health facility, previously identified as a strong determinant of child mortality and correlated with wealth was included as an adjustment variable in the sensitivity and adjusted models.

For the wealth-AZ interaction models, we have clarified (line 249) that we used Poisson regression models similar to those described above, including both the main effects of wealth and AZ, as well as their interaction term to assess interaction on a multiplicative scale.

We also expanded the description of the PCA approach (line 194) as follows: “PCA was used to generate the wealth index because it efficiently summarizes correlated asset variables into a single composite score and is widely used for constructing asset-based wealth indices in demographic and health studies, including those conducted in sub-Saharan Africa.”

This approach captures the largest proportion of variability in socioeconomic status and maintains comparability with DHS and other population-based surveys. Factor analysis, while conceptually similar, assumes underlying latent constructs and requires additional distributional assumptions that were not necessary for the purpose of deriving an empirical wealth index.

5. Minor:

Results — Table 3: Emphasize or clearly mark the statistically significant values.

Response: Thank you for the suggestion. All estimates presented in Table 3 were statistically significant at α = 0.05 (P < 0.05). We have added a note below the table to clarify this convention.

We wish to reiterate our gratitude for being given the opportunity to respond to the Reviewers’ comments as well as for being considered for publication in PLOS One Journal. If there are any further questions or comments, please do not hesitate to contact me.

Sincerely,

Elisabeth Gebreegziabher, PhD, MPH

---

## [Decision Letter · Decision Letter 2]

12 Jan 2026

Exploring Heterogeneity in Treatment Effects: The Impact and Interaction of Asset-Based Wealth and Mass Azithromycin Distribution on Child Mortality

PONE-D-25-23672R2

Dear Dr. Gebreegziabher,

We’re pleased to inform you that your manuscript has been judged scientifically suitable for publication and will be formally accepted for publication once it meets all outstanding technical requirements.

Kind regards,

Tsegaye G. Haile

Academic Editor

PLOS One

Additional Editor Comments (optional):

Reviewers' comments:

Reviewer's Responses to Questions

**Comments to the Author**

Reviewer #1: All comments have been addressed

Reviewer #3: All comments have been addressed

2. Is the manuscript technically sound, and do the data support the conclusions?

Reviewer #1: Yes

Reviewer #3: Yes

3. Has the statistical analysis been performed appropriately and rigorously?

Reviewer #1: Yes

Reviewer #3: Yes

4. Have the authors made all data underlying the findings in their manuscript fully available?

Reviewer #1: Yes

Reviewer #3: Yes

5. Is the manuscript presented in an intelligible fashion and written in standard English?

Reviewer #1: Yes

Reviewer #3: Yes

Reviewer #1: I am satisfied with the authors' responses and the resulting changes to the paper. I have nothing else to add.

Reviewer #3: Thank you very much for the revised version of the manuscript. All comments have been well addressed.

**Do you want your identity to be public for this peer review?** For information about this choice, including consent withdrawal, please see our Privacy Policy

Reviewer #1: No

Reviewer #3: **Yes:** EE VIEN LOW

---

## [Editor Report · Acceptance letter]

PONE-D-25-23672R2

PLOS One

Dear Dr. Gebreegziabher,

I'm pleased to inform you that your manuscript has been deemed suitable for publication in PLOS One. Congratulations! Your manuscript is now being handed over to our production team.

Kind regards,

on behalf of

Mr. Tsegaye G. Haile

Academic Editor

PLOS One